# Within-Host Adaptation of *Staphylococcus aureus* in a Bovine Mastitis Infection Is Associated with Increased Cytotoxicity

**DOI:** 10.3390/ijms22168840

**Published:** 2021-08-17

**Authors:** Katharina Mayer, Martin Kucklick, Helene Marbach, Monika Ehling-Schulz, Susanne Engelmann, Tom Grunert

**Affiliations:** 1Functional Microbiology, Institute of Microbiology, Department of Pathobiology, University of Veterinary Medicine, 1210 Vienna, Austria; katharina.mayer@vetmeduni.ac.at (K.M.); helene.marbach@vetmeduni.ac.at (H.M.); monika.ehling-schulz@vetmeduni.ac.at (M.E.-S.); 2Institute for Microbiology, University of Technical Sciences, 38106 Braunschweig, Germany; martin.kucklick@helmholtz-hzi.de (M.K.); susanne.engelmann@helmholtz-hzi.de (S.E.); 3Microbial Proteomics, Helmholtz Centre for Infection Research, 38124 Braunschweig, Germany

**Keywords:** within-host adaptation, in-host evolution, *Staphylococcus aureus*, extracellular persistence, chronic bovine mastitis, cytotoxicity, SigB-deficiency

## Abstract

Within-host adaptation is a typical feature of chronic, persistent *Staphylococcus aureus* infections. Research projects addressing adaptive changes due to bacterial in-host evolution increase our understanding of the pathogen’s strategies to survive and persist for a long time in various hosts such as human and bovine. In this study, we investigated the adaptive processes of *S. aureus* during chronic, persistent bovine mastitis using a previously isolated isogenic strain pair from a dairy cow with chronic, subclinical mastitis, in which the last variant (host-adapted, Sigma factor SigB-deficient) quickly replaced the initial, dominant variant. The strain pair was cultivated under specific in vitro infection-relevant growth-limiting conditions (iron-depleted RPMI under oxygen limitation). We used a combinatory approach of surfaceomics, molecular spectroscopic fingerprinting and in vitro phenotypic assays. Cellular cytotoxicity assays using red blood cells and bovine mammary epithelial cells (MAC-T) revealed changes towards a more cytotoxic phenotype in the host-adapted isolate with an increased alpha-hemolysin (α-toxin) secretion, suggesting an improved capacity to penetrate and disseminate the udder tissue. Our results foster the hypothesis that within-host evolved SigB-deficiency favours extracellular persistence in *S. aureus* infections. Here, we provide new insights into one possible adaptive strategy employed by *S. aureus* during chronic, bovine mastitis, and we emphasise the need to analyse genotype–phenotype associations under different infection-relevant growth conditions.

## 1. Introduction

Bovine mastitis is the inflammation of the mammary gland and udder tissue mainly induced by microbial pathogens. Not only does mastitis cause the worst disease-related economic losses for the dairy industry, but it also leads to reduced animal welfare and food safety [1]. The Gram-positive bacterium *Staphylococcus aureus* is one of the most common causes of contagious bovine mastitis. Current antimicrobial treatments and vaccination against *S. aureus* bovine mastitis are not very effective, and in most cases, a predominant *S. aureus* clone persists for long periods within its host [2]. These persistent, recurrent infections are mostly subclinical, with few, if any, symptoms in the animal, but causing a loss in milk yield and quality as well as being a reservoir for the pathogen to spread within the herd. Extracellular biofilm formation and intracellular survival have been linked to staphylococcal escape from the host’s immune system in bovine mastitis [3,4]. However, there is still a largely unknown gap regarding the underlying complexity of the host-pathogen interaction that allows long-term persistence within the animal and the spread of *S. aureus* in herds.

Surface-associated and secreted proteins and surface polysaccharides are key *S. aureus* virulence determinants [5,6]. In bovine mastitis, these factors are involved in numerous functions, particularly adhesion to the extracellular matrix and host cells, invasion of host cells, evasion of the host’s immune responses and biofilm formation. Their direct contact with the udder microenvironment allows for permanent, selective and functional changes as a host-adaptive process. Adaptation to the bovine udder has been associated with distinct genetic and phenotypic traits in *S. aureus*, including genes involved in virulence and cell envelope composition [7,8]. Bardiau et al. proposed that certain groups of bovine *S. aureus*, equipped with specific phenotypic features, are more likely adapted to either the intra- or extracellular niche within the bovine host [9,10]. We recently described two coexisting persistent *S. aureus* subtypes, differing in their within-herd prevalence and exhibiting a distinct phenotypic trait profile [8]. High within-herd prevalence was associated with lack of capsular polysaccharide (CP) expression, high cellular invasiveness, low cytotoxicity and high poly-*N*-acetylglucosamine (PNAG)-based biofilm production. These results suggest that the capacity of *S. aureus* to cause bovine mastitis may not only rely on traits specifically associated with the bovine host but also on niche-specific *S. aureus* adaptations to the local microenvironment within the bovine udder.

The present study investigated adaptive bacterial processes under in vitro growth-limiting conditions that bacteria face during infections associated with the bovine udder cellular micro-environment. In particular, we examined the effect of combined stress situations (nutrient, iron and oxygen limitation using RPMI), which were shown to induce in vivo-like staphylococcal surface antigens [11,12]. We analysed a pair of isolates collected from a single naturally infected dairy cow with chronic, subclinical mastitis. Genetic and phenotypic characteristics of the initial (IN, first isolate) and the host-adapted isolate (HA, isolated after three months) have already been investigated and showed the transition towards a SigB-deficient phenotype characterised by reduced CP expression but enhanced proteolytic activity and PNAG-based biofilm production determined under aerobic, nutrient-rich conditions [13]. The isogenic strain pair was comparatively assessed regarding their surface protein and cell envelope glycan composition using mass spectrometry- (MS) and Fourier-transform infrared (FTIR) spectroscopy-based techniques, respectively. Phenotypic properties (including cellular lysis and biofilm assays) and the expression of defined surface and secreted virulence factors (including CP, PNAG and alpha-hemolysin) were further determined to get better insights into adaptive traits relevant under a specific infection-relevant condition within the bovine udder.

## 2. Results

### 2.1. Differences in the Surface Proteome between IN and HA

We compared the cell surface-associated proteome of the initial (IN) and host-adapted (HA) isolate under infection-relevant growth conditions to investigate changes due to host-adaptive processes of *S. aureus* to the udder niche during bovine chronic mastitis. Strains were grown in iron-depleted RPMI under oxygen limitation (from now on referred to as RPMI), harvested in the early stationary phase (10 h) (Appendix A), and the bacterial surface-associated proteome was investigated using an MS-based trypsin shaving and surface biotinylation approach. In total, 498 and 352 proteins were identified by either the shaving or biotinylation approach; 240 proteins were found with both methods (Figure 1). Notably, only 6 proteins were detected in significant different quantities in the IN and HA isolates, suggesting a limited impact of host-adaptive conditions on the bacterial surfaceome (Appendix A). The most prominent decreases were observed for the surface-associated lipoprotein Newbould305_0930 (EJE57358.1) and the membrane-associated Asp23 protein (EJE56063.1; 5.4- and 12.0-fold change, respectively; *p* < 0.001) in HA. The other four proteins found are predicted to be cytosolic or membrane-associated and can be assigned to various functional categories involved in carbohydrate (HPr(Ser)kinase phosphatase, Hrpk; Gluconate operon transcriptional repressor, Gntr) and nucleotide (Ribonucleotide-diphosphate reductase subunit beta, NrdF) metabolism, or antibiotic resistance (hypothetical protein Newbould305_1521 identified by protein BLAST search as glycopeptide resistance-associated protein GraF).

### 2.2. Differences in the Cell Envelope Glycopolymer Composition between IN and HA

Glycopolymers are essential components of the staphylococcal cell envelope and play critical roles in host cell interactions, immune evasion and persistence [14]. To obtain an overview of adaptive alterations in staphylococcal surface glycopolymers during bovine mastitis, we employed Fourier-transform infrared (FTIR) spectroscopy, a method suitable to determine general changes in the cell envelope glycan composition including CP and wall teichoic acid (WTA) glycoepitopes [15,16]. IN and HA were grown under the same conditions used in the proteomics approach (RPMI under iron- and oxygen-limiting conditions), and spectral data were recorded at seven different time points of bacterial growth (Appendix A).

FTIR spectral analysis revealed differences between IN and HA of staphylococcal surface glycopolymer expression in the polysaccharide region (1200–800 cm^−1^) (Figure 2). In particular, growth phase-independent differences between IN and HA in the spectral region between 1080 and 880 cm^−1^ were identified. Furthermore, spectral differences at wavenumbers 1100 cm^−1^/1085 cm^−1^ were either observed in the exponential phase or at wavenumbers 850 cm^−1^/833 cm^−1^ at the stationary phase. Growth phase-independent differences between IN and HA can be attributed to stretching vibrations of sugar ring structures at the staphylococcal cell envelope [17]. Furthermore, alterations at the exponential phase are mainly associated with the vibration of functional groups (C-O-C, C-O, C-O-P, P-O-P, P=O) of phosphate-containing cell envelope-associated polysaccharides [18,19], and in the stationary phase to changes in the anomeric configuration of the glycosidic linkage type of polysaccharides [20].

Next, we attempted to clarify the differences in glycopolymers obtained by FTIR spectroscopy between IN and HA, determining the surface expression of CP and PNAG using immune-based slot blot assays. Our previous study revealed an increased CP expression for the IN and increased PNAG content in biofilms for the HA isolate grown under aerobic, nutrient-rich (tryptone-soy base) conditions [13]. However, neither CP nor PNAG was detected at IN and HA’s bacterial surface under the employed growth conditions (Figure 3). Thus, the differences in staphylococcal cell envelope glycopolymers determined by FTIR spectroscopy cannot be linked to differences in CP or PNAG but can be attributed to yet-to-be-explored differences in other cell envelope carbohydrate moieties such as wall-/lipo-teichoic acids (WTA/LTA) and peptidoglycan (PG) [16].

Further, the two isolates were tested for their capacity to produce biofilm and PNAG in biofilm using a microtiter plate biofilm assay. In contrast to our previous study conducted under standard laboratory conditions in TSB [13], the isolates grown under these infection-relevant conditions showed no difference in biofilm formation and PNAG production (Appendix A).

### 2.3. Increased Cytotoxicity of HA

We tested for increased alpha-hemolysin (Hla, α-toxin) production of HA in RPMI, as this has been described as a feature of SigB-deficient strains [21]. Bacterial supernatants were analysed for secreted Hla (33 kDa) by western blot analyses. As expected, the antibody-based detection revealed a higher Hla protein concentration for the HA isolate in the supernatant. Only a weak protein band was detected for IN compared to a much more intense band for HA (Figure 4A).

Thus, we subsequently tested the in vitro cytotoxicity of the two isolates using red blood cells as wells as bovine mammary epithelial cells (MAC-T). For the haemolysis assay, sheep erythrocytes were co-cultured with bacterial supernatant for 60 min at 37 °C, and haemolysis was assessed by measuring the amount of free haemoglobin released by damaged red blood cells. Measurement of free haemoglobin revealed a significantly stronger haemolytic activity for the HA isolate (*p* < 0.041) (Figure 4B). MAC-T cells were incubated for 24 h with bacterial supernatant and analysed for cell viability to examine the cytotoxic potential of the isogenic strain pair on bovine mammary epithelial cells. In line with the immunoassays targeting larger quantities of Hla, we detected a significant (*p* < 0.001) increased cytotoxicity on MAC-T cells for HA (Figure 4C).

## 3. Discussion

*S. aureus* can infect multiple host species and survive in diverse ecological niches [22]. During long-term chronic infections, *S. aureus* is exposed to the host’s immune system and changing nutrient sources, thus promoting adaptation of the pathogen to the host niche-specific environment. We recently reported within-host *S. aureus* evolution during a chronic, subclinical bovine mastitis associated with a single nucleotide polymorphism in *rsbU* (G368A→G122D), an activator of the alternative sigma factor SigB [13]. SigB is involved in controlling *S. aureus* stress response and virulence factor expression [23]. SigB-deficient mutants were frequently isolated from human infections [24,25], and a high frequency of mutations in SigB-associated loci was reported for *S. aureus* [26]; however, as highlighted by our recent work, this loci might indeed represent a vital region prone to accumulate genetic variations even in the non-human host. In the present study, we aimed to investigate how these changes in the staphylococcal genome result in phenotypic adaptations that potentially allow the pathogen to better survive under infection-relevant, host cell/tissue-like growth conditions. For this purpose, we comparatively assessed the initial (IN) and host-adapted (HA) isolate grown in iron-depleted RPMI at 37 °C under oxygen-limited conditions, which was shown to sufficiently induce in vivo-like staphylococcal antigen production during ruminant mastitis as examined by serological proteome analysis (SERPA) [11,12].

RPMI 1640 medium is frequently used to cultivate eukaryotic cells, such as endothelial cells and professional phagocytes (e.g., macrophages) and had already been used to simulate *S. aureus* growth in plasma [27]. Limited iron availability was shown to increase the expression of bacterial virulence factors [28,29,30] and led to the identification of several immuno-dominant proteins by SERPA derived from bovine *S. aureus* strains [31]. Oxygen-limiting conditions were applied because the oxygen concentration drops remarkably (below 10%) in milk from cows with mastitis [32].

We conducted a complementary approach of the *S. aureus* cell surface-associated proteome between IN and HA using biotinylation and trypsin shaving. Despite the many proteins identified, we found only six proteins significantly differentially abundant between IN and HA. Among them, the SigB-dependent Asp23 protein (homolog to SACOL2173 in strain *S. aureus* COL) was less present in HA [33,34]. Asp23 is described to be one of the most abundant proteins in *S. aureus* [35]. Asp23 is a membrane-associated protein, and *asp23* mutant strains showed an increased cell wall stress response suggesting a pivotal role in staphylococcal cell envelope homoeostasis [36]. It is transcriptionally regulated by the alternative sigma factor SigB, controlling *S. aureus* stress response and virulence gene expression [23]. Likewise, the less abundant lipid-anchored protein Newbould305_0930 (homolog to SACOL0444) was described to be strongly SigB-dependent, as reported by a cell-surface proteomic-profiling approach [34]. The remaining four proteins were assigned to different functional categories or are hypothetical proteins predicted to be mainly cytosolic. These cytosolic proteins have been described to be involved in various cellular biochemical processes. Since we detected these proteins at the cellular surface, they might exert additional functions in bacterial pathogenesis known as moonlighting function [37]. However, we found no evidence for this in the literature. Notably, we identified, by protein BLAST search, the hypothetical protein Newbould305_1521 (decreased in HA) as glycopeptide resistance-associated protein GraF. When upregulating *graF* promoter activity, *S. aureus* was shown to be less susceptible to vancomycin, teicoplanin and oxacillin [38]. In agreement with the findings of our previous study, by Marbach et al. [13], we identified two proteins (Asp23 and SACOL0444) controlled by the SigB regulon [23,34]. In contrast to our study, Hempel et al. identified 49 surface-associated, SigB-dependent proteins [34], which can partially be explained by the differently used growth conditions and partially by the more stringent settings for data evaluation. For instance, applying the less stringent criteria of Hempel et al. to our MS dataset, the following proteins, amongst others, would additionally be scored as less abundant in HA: membrane-anchored Ser-Asp rich fibrinogen-binding protein (SdhR); and scored as higher abundant in HA: gamma-hemolysin component A (HlgA), beta-hemolysin (Hlb), leukocidin LukM, leucocidin LukS, UDP-N-acetylmuramoylpentapeptide-glycine glycosyltransferase (FemA), staphopain cysteine proteinase (SspB) and staphylococcal thermonuclease (Nuc). In particular, Hlb and the detected subunits of bi-component leukotoxins (HlgA, LukM, LukS) [39,40] are associated with cytotoxic effects.

Indeed, we found the most significant differences between IN and HA associated with their cytotoxic phenotype. We detected increased cytotoxicity on MAC-T cells and red blood cells of HA. We also obtained an increased abundance of secreted Hla in the RPMI culture supernatant in HA, contributing to the cytotoxic and hemolytic effects. Possibly, other hemolysins (HlgA, Hlb) or leukocidines (LukM, LukS), as revealed by our proteomics approach, may contribute to the higher hemolytic and cytotoxic activity for HA. Elevated Hla expression and secretion was expected for HA because it is typical for SigB-deficiency [21]. The pore-forming toxin Hla significantly facilitates immune evasion by damaging phagocytic cells and improving penetration to distant tissues [41]. In bovine mastitis, a high Hla toxin expression in vitro correlates with more severe tissue necrosis during experimental mastitis infection [42]. Despite the more cellular destructive effects, a more severe infection cannot be assumed, supported by the significantly lower mortality that we detected for HA in a mouse infection model [13].

Yet, the role of Hla-mediated cytotoxicity in *S. aureus* pathogenicity in bovine mastitis is still under discussion. On the one hand, low-cytotoxicity strains were proposed to represent a source for chronic infections [43,44]. Supporting this, *S. aureus* Δ*hla* mutant strains caused less severe infections in a mouse model of intramammary infection [45]. On the other hand, Monecke et al. did not find a correlation between in vitro Hla toxin production and the severity of mastitis infections [46]. Furthermore, our field studies revealed that low- and high-cytotoxic strains could persist in parallel for years in a dairy herd, although the high-cytotoxic strains had a lower within-herd prevalence [8]. In addition, Genotype B (GTB/ST8) bovine strains with a high propensity to cause chronic mastitis infection were associated with higher potential cytotoxicity [47,48]. Thus, the contribution of cytotoxicity in chronic, long-lasting *S. aureus* infection appears to be highly complex and suggests different strategies for the long-term survival of this pathogen in the bovine host. Indeed, the pronounced Hla secretion of the SigB-deficient HA does not support the hypothesis for prolonged intracellular survival within the cells of the infected mammary gland. To support this, intracellular survival requires a functional SigB to form small colony variants (SCV’s) [49]. Thus, SigB-deficient strains may employ an alternative strategy of long-term survival, presumably facilitating extracellular persistence within the host. Determined by the local conditions in the mammary gland tissue microenvironment, such as nutrient and oxygen limitation, it is tempting to speculate that increased cytotoxicity may facilitate immune evasion and bacterial dissemination. The switch to the more “invasive/aggressive” phenotype could help to extract nutrients from tissue to overcome this limitation temporarily. Thus, the increased ability to subvert local mammary gland defences and exploit new nutritional sources restricted to a particular stage of infection could improve subclinical, chronic *S. aureus* mastitis. Indeed, future work is needed to fully understand the interplay between cytotoxicity and *S. aureus* persistence in dairy cattle.

Unexpectedly, we detected no differences between IN and HA in RPMI for several phenotypic traits previously shown to be significantly increased in HA in TSB media, including surface PNAG expression and PNAG-based biofilm production [13]. In particular, the observed switch from CP expression in IN to PNAG expression in HA when grown in TSB was not evident in RPMI, where neither of the staphylococcal surface glycopolymers was detected. *S. aureus* produces CP and PNAG from the same biosynthetic precursor (UDP-*N*-acetylglucosamine), although either derived from gluconeogenesis or glycolysis, respectively [50]. *S. aureus* fine-tunes CP and PNAG production by coordinating multiple inputs, including available nutrient levels [51,52]. Thus, we assume that *S. aureus* expression of costly, surface glycopolymeric virulence factors during nutrient and oxygen-limiting conditions is diminished, perhaps due to the lack of available UDP-*N*-acetylglucosamine. Possible implications for developing vaccines based on CP/PNAG components are indicated if these are not produced at all under certain conditions. Moreover, we observed a shift from changes in phosphate-containing functional groups to stereochemical changes in sugar molecules during bacterial growth by FTIR spectroscopy, which sounds plausible, reflecting the chronological sequence in cell envelope assembly from phosphodiester-linked glycerol-phosphate (GroP) or ribitol-phosphate (RboP) subunits of teichoic acids towards adding α/β-GlcNAc substituents. Further investigations in the glycostructural composition are necessary to dissect differences in the carbohydrate-associated spectroscopic signature between IN and HA obtained by FTIR spectroscopy, which could neither be related to CP nor PNAG.

This study presents two limitations, which should be addressed in future research. First, our applied in vitro culture conditions in RPMI simulating combined stress situations (nutrient, iron and oxygen limitation) may therefore reflect only partially the conditions of the infected mammary gland in the complex staphylococcal lifecycle during bovine mastitis. Nevertheless, these conditions have been shown to induce in vivo-like staphylococcal surface antigens during ruminant mastitis, suggesting they display conditions at least for specific host niches [11]. Since RPMI represents a low-nutrient/oxygen-deprived environment, it most likely reflects the conditions of the cellular microenvironment in deeper layers of the udder tissue. Thus, our findings might contribute to the understanding of mechanisms of persistence in these rather local conditions within the bovine mammary gland, instead of explaining the general mechanism of *S. aureus* persistence in bovine mastitis. Second, when interpreting the results of this study, which relates to a single case of within-host adaptation in *S. aureus* bovine mastitis, it must be considered that the findings are linked to the specific bacterial genetic background and selective pressure of the individual dairy cow and therefore cannot be extrapolated to a common adaptive strategy of bovine *S. aureus*. Considering these limitations of our study, as mentioned above, we hypothesise that the high cytotoxic potential of HA will possibly gain an advantage if the bacteria comes under pressure within the bovine host. However, we cannot exclude the possibility that other yet unknown SigB-dependent factors, such as metabolic adaptations, provide HA with an advantage over IN. Further investigations should be undertaken to explore SigB-deficient adaptive traits using a larger strain collection and employing additional growth conditions reflecting the different environments that bacteria encounter in the udder niche, including milk proteins and lactose.

## 4. Materials and Methods

### 4.1. Bovine Isolates and Reference Strains

Bovine *S. aureus* isolates were isolated and initially characterised, as described elsewhere [13]. Briefly, the isogenic strain pair, initial isolate (IN, collected at the first sampling point) and host-adapted isolate (HA, isolated three months later) were recovered from milk samples following *S. aureus* within-host adaptation during the progression in chronic, bovine mastitis. We conducted a high-frequency, at least weekly, sampling during the 3-months period of the study and analysed the isolates derived from the same udder (n = 21) by spectroscopic fingerprinting to detect possible phenotypic changes, supplemented by whole-genome sequencing to prove clonal ancestry of the isolates [13]. Moreover, during our study, the cow was kept separately in the University’s intern cowshed, and milking was performed with separate milking equipment, following usual hygiene practices, thus preventing possible transmission events to other cows as well as possible re-infections. *S. aureus* laboratory strains 6850 [53], and 6850Δ*hla* (kindly provided by Lorena Tuchscherr de Hauschopp, Institute of Medical Microbiology, University Hospital of Jena, Germany) were used as reference strains in cytotoxicity assays and alpha-toxin (Hla) western blot analysis. Strain Reynolds CP5 and Reynolds CP- (non-encapsulated) [54] were used as assay controls for CP detection. Strain 113 and 113Δ*ica* [55] were used as controls for PNAG assessment.

### 4.2. Bacterial Growth under the Infection-Relevant Conditions

Bacterial isolates were grown under iron- and oxygen-limiting conditions in RPMI 1640 medium, earlier described to be best mimicking growth in vivo during mastitis [11]. Bacteria were cultivated overnight in RPMI 1640 medium supplemented with 0.15 mM deferoxamine (DFOM) (Sigma-Aldrich, St. Louis, MO, USA) and then diluted to OD_600_ of 0.05 in the same medium. Strains were cultivated under restricted oxygen conditions in sealed tubes completely filled with media, without agitation at 37 °C.

### 4.3. MS-Based Surface Proteomics

For the proteomic surface-shaving and a surface-biotinylation approach, samples were harvested by centrifugation at 10 h (early stationary phase) (Appendix A). Bacterial cells were washed once with PBS and collected by centrifugation (5 min, 4 °C, 4000× *g*). Each strain was investigated in three independent experiments. Bacterial surface-biotinylation and trypsin shaving approaches using LC-MS/MS are described in detail in the Appendix A (Appendix A and Methods).

### 4.4. FTIR Spectroscopic Measurement and Analysis

Samples were harvested by centrifugation (8000× *g*, 20 min, RT) from 6 to 18 h in two-hour intervals (Appendix A). The pellet was washed once with 1 mL PBS (5000× *g*, 2 min, RT) and resuspended in 100 μL of sterile deionised water. Subsequently, 30 μL each of the bacterial suspensions were spotted on a zinc selenite (ZnSe) optical plate, dried at 40 °C for 40 min, and measured with an HTS-XT microplate adapter coupled to a Tensor 27 FTIR spectrometer (Bruker Optics GmbH, Ettlingen, Germany). OPUS software (version 6.5; Bruker Optics GmbH) was used to process the measured FTIR spectra and perform chemometric analysis. Subtraction spectra were generated from second derivative, vector-normalised, average FTIR spectra by subtracting HA spectra from IN spectra for each time point. For a hierarchical cluster analysis (HCA), the spectral region that offers information about carbohydrate constituents (1200–800 cm^−1^) was selected, which was shown to be highly discriminatory, based on the expression of CP and/or other cell surface glycans, such as wall teichoic acid [16,56]. Bacterial strains were grown three times independently, each with three technical replicate measurements.

### 4.5. Immune-Based Detection of Glycopolymers and Hla

The expression of the *S. aureus* surface-associated glycopolymers, CP and PNAG, were performed by slot blot immune-based detection. Bacteria were harvested after 10 h and 18 h growth in iron-depleted RPMI medium by centrifugation (5 min, 4 °C, 4000× *g*). Capsule extracts were produced and processed, as described earlier [57]. Briefly, capsule extract dilutions of 1:100 were loaded on the slot blot and subsequently detected with a polyclonal anti-CP 5 (serotype 5) antibody (kindly provided by Fernanda R. Buzzola, Universidad de Buenos Aires, Argentina) and visualised by the protein-A-HRP (P8651, Sigma-Aldrich) [58]. IN and HA were also grown on tryptone soy agar (TSA, Oxoid Limited, Hampshire, UK) along the control strains, Reynolds CP5 and Reynolds CP-, and further processed in the same way.

PNAG extract preparation and slot blot procedure were performed as previously described [59]. Diluted samples of 1:10 were detected by a monoclonal human anti-PNAG antibody (kindly provided by Gerald B. Pier, Harvard Medical School, Boston, MA, USA) and visualised by a goat anti-human IgG-HRP antibody (CATNO 2042-05, Southern Biotech, Birmingham, AL, USA). The strains 113 and 113Δ*ica*, grown on TSA, were used as controls.

Hla western blots were performed according to Grunert et al. [60] with slight modifications. Detection was achieved by the combination of the anti-alpha-hemolysin antibody [8B7]—N-terminal (#ab190467) (Abcam, Cambridge, UK) diluted 1:3000, and the peroxidase-conjugated AffiniPure Goat Anti- Mouse IgG (H + L) antibody (#115-035-062) (Dianova, Hamburg, Germany) diluted 1:20,000. Hla proteins bands were detected by Clarity Max Western ECL Substrate (BioRad, Hercules, CA, USA).

### 4.6. Biofilm Formation and PNAG-Production

The amount of PNAG production and biofilm formation was analysed, as described previously [8]. Briefly, bacteria were grown under static and oxygen limiting conditions in 96-well microtiter plates for 24 h at 37 °C without agitation. The biofilm was fixed with methanol followed by 0.1% crystal violet staining, and the absorbance was measured at 590 nm. The PNAG content was visualised at OD 490 nm using a wheat-germ agglutinin horseradish peroxidase conjugate (WGA–HRP, 30 min at 37 °C) (Sigma-Aldrich, St. Louis, MO, USA) and o-phenylenediamine (OPD) as substrate.

### 4.7. Cytotoxicity and Hemolysis Assays

The isolates’ cytotoxic potential against mammary epithelial cells was investigated using a cell culture assay described earlier [8]. Briefly, MAC-T cells [61] were incubated with sterile filtered bacterial supernatant for 24 h to determine the remaining cell viability based on the reduction of tetrazolium salt WST-1 (Roche, Basel, Switzerland) by the mitochondrial activity of viable cells.

Hemolytic activity testing was performed similar to that described by Hirschhausen et al. [62]. The supernatant of bacterial culture (10 h) was harvested by centrifugation and then sterile filtered. An amount of 100 µL of undiluted supernatant was incubated with 100 µL of a 10% (*v*/*v*) working solution of purified sheep erythrocytes (Fiebig Nährstofftechnik, Idstein-Niederauroff, Germany).

The laboratory strains 6850 and 6850Δ*hla* were used as references in both assays.

### 4.8. Statistics

Statistical analysis was performed using the GraphPad Prism 7.0 software. Differences between the IN and HA isolate were tested by an unpaired, two-tailed Students *t*-test (Biofilm-PNAG) and non-parametric, two-tailed Mann–Whitney U tests (Biofilm formation; cytotoxicity; hemolysis assays). Statistical significance was described by * *p <* 0.05; ** *p* < 0.01; *** *p* < 0.001.

## 5. Conclusions

Our data indicate that the *S. aureus* SigB-deficiency evolved during in-host adaptation in chronic, persistent bovine mastitis is associated with increased cytotoxicity when grown in vitro under a specific infection-relevant growth-limiting condition. This supports our previous assumption that SigB-deficiency most likely facilitates extracellular persistence within the host [13]. Moreover, we provide further evidence that SigB is crucial in triggering responses to environmental signals and directs the adaptive expression of virulence genes. Depending on the pathogens’ niche within the bovine udder, adaptations can differ during persistence. Localised in a relatively nutrient-depleted microenvironment, it will be advantageous to secrete more Hla to lyse cells to be able to penetrate and disseminate in the udder tissue, possibly evading host immune defences and exploring new nutritional sources. Here, we demonstrate that *S. aureus* does not express costly, surface-associated glycopolymers, which may affect CP/PNAG-targeted vaccines’ efficacy if immunogens under the used growth conditions are not produced. It remains to investigate how adaptive traits are affected by simulating alternative extracellular host-like growth environments such as milk. We further emphasise the need to establish in vitro models with growth conditions representing various udder-related infection sites to study the bacterial pathogenicity more temporally and spatially during bovine mastitis infection.

## Figures and Tables

**Figure 1 ijms-22-08840-f001:**
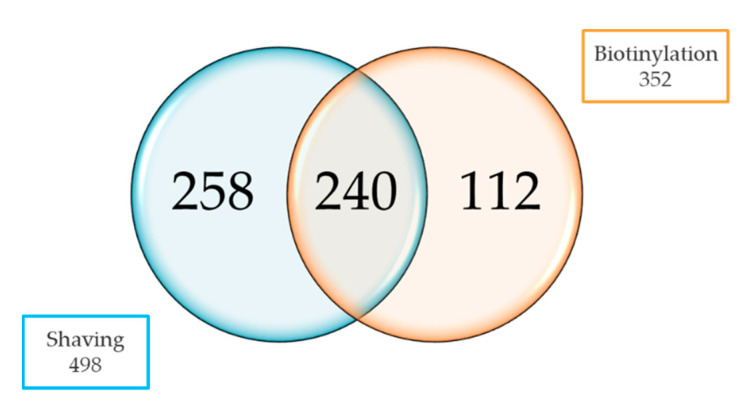
The number of identified proteins using surface proteomics. Venn diagram illustrating the numbers of identified proteins of the two complementary cell surface proteomics approaches, trypsin-shaving approach (blue) and biotinylation approach (orange).

**Figure 2 ijms-22-08840-f002:**
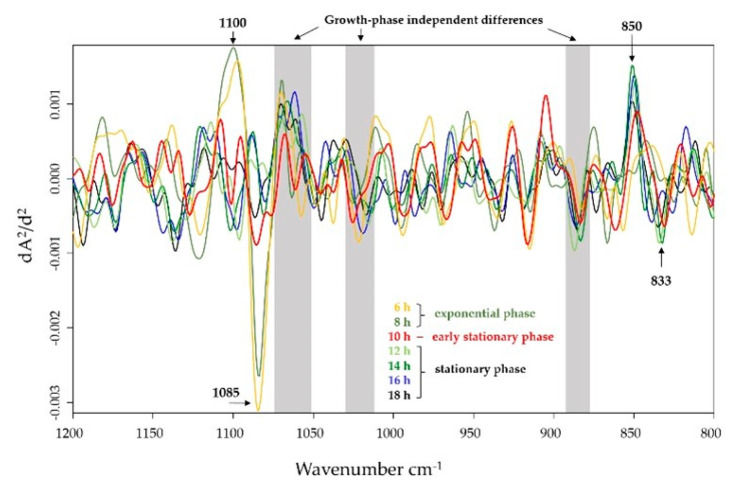
Differences between IN and HA surface-associated glycopolymer expression were examined by FTIR spectroscopy. Alteration in the expression of *S. aureus* surface-associated glycopolymers of IN and HA was monitored between 6 and 18 h by 2 h intervals. Spectral comparison of bacteria at different time points was performed using the polysaccharide spectral region (1200–800 cm^−1^), dominated by vibrations of various carbohydrates and their specific types of glycosidic linkages. Subtraction spectra were generated from the second derivative, vector-normalised, average *S. aureus* FTIR spectra. Spectra from the HA isolate were subtracted from that of the IN isolate.

**Figure 3 ijms-22-08840-f003:**
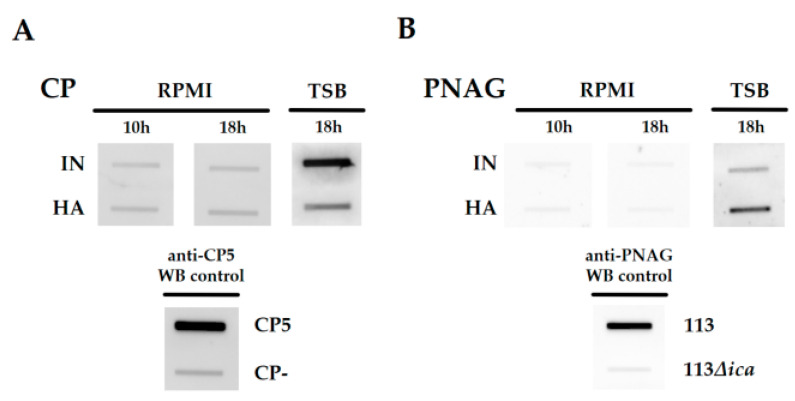
Immune-based detection of CP and PNAG. Slot blot analysis of CP and PNAG after 10 h growth in RPMI and 18 h growth in RPMI and TSB, including control strains. (**A**) CP extracts of IN, HA and the control strains Reynolds CP5 and CP- were used 1:100 diluted for the analysis. (**B**) All PNAG extracts (IN, HA, 113, 113Δ*ica*) were used 1:10 diluted for the antibody-based detection (**A**,**B**). Data are representatives of three independent experiments. IN, initial isolate; HA, host-adapted isolate.

**Figure 4 ijms-22-08840-f004:**
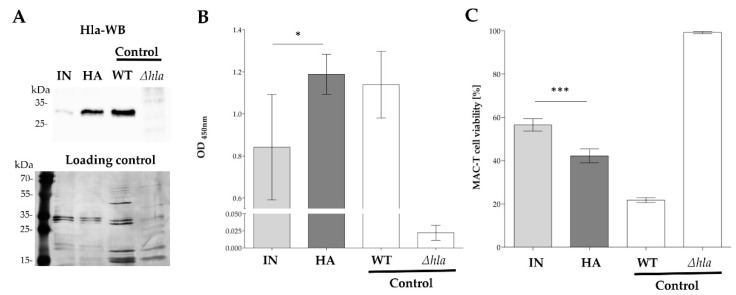
Hla detection, hemolysis and cellular cytotoxicity. (**A**) Western blot of alpha-hemolysin. IN and HA isolates were cultivated 24 h in RPMI, supernatants were collected, concentrated and analysed for their Hla content. A silver-stained SDS-gel was used as a loading control to ensure equal protein quantities per lane. Data are representatives of three independent experiments. (**B**) Hemolytic activity was determined using absorbance at 450 nm (released haemoglobin from sheep red blood cells after incubation with undiluted supernatant for 60 min at 37 °C). The bars represent the mean and 95% CI of three independent biological replicates and two technical replicates. (**C**) Cytotoxic effect of bacterial culture supernatant on a bovine mammary epithelial cell line (MAC-T cells). Cells were incubated with undiluted bacterial supernatants for 24 h to analyse the cell viability. Measurements were normalised to the cytotoxicity of MAC-T cells treated with RPMI + DFOM medium. Experiments were performed in three independent biological replicates, each with three technical replicates +/− 95% CI. (**A**–**C**) (* *p* < 0.05; *** *p* < 0.001). IN, initial isolate; HA, host-adapted isolate. Assay control strains: WT, Strain 6850; Δ*hla*, Strain 6850Δ*hla*.

## Data Availability

The mass spectrometry proteomics data have been deposited to the ProteomeXchange Consortium (http://proteomecentral.proteomexchange.org) via the PRIDE [67] partner repository with the dataset identifier PXD025013 (https://www.ebi.ac.uk/pride/archive).

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
