# Peer review of "Within-Host Adaptation of Staphylococcus aureus in a Bovine Mastitis Infection Is Associated with Increased Cytotoxicity"

_ijms, 2021, doi:10.3390/ijms22168840_

Round 1

Reviewer 1 Report

Comments on the article Exploring New Niches: Within-Host Adaptive Responses of Bovine Staphylococcus Aureus Are Associated with Increased Cytotoxicity

The article explores the mechanisms involved in the adaptation of Bovine S.a adaptations isolated from cows with persistent mastitis.

This pathogen is one of the most important diseases in the dairy industry, and therefore, any specific research is always welcome.

The article approach is interesting, but I believe that methodology could be biasing the results

Among them, the authors used one strain from the same cow (I understand that), within 3 months of difference. Two important concerns arise.

First, I wondering how the authors can be sure that HA is adapted and it is not caused by a re-infection.

Second and more important, It is highly difficult to characterize Staph aureus (as stated in the title) using a single individual as the source of bacterial samples.

In addition, the results presented in this study are contradictory (an even switched) with the results provided by the same group before (Acknowledged in L 280). Despite the author's hypothesis regarding the cause, it could be produced by the use of an alternative culture media in comparison with the former reports. Despite that authors used the new media to “mimic” the natural conditions of the mammary gland, one of the results is biased (this one or the previous) by the experimental culture conditions. In any case, the results presented in this study are far to be reliable to hypothesize regarding the general mechanism, and therefore, they should be narrowed to the specific culture conditions rather than to the “disease” in general.

My suggestion is that the authors re-calibrate the aim (and the title) of the study to a more realistic condition and avoid concluding beyond the results presented here.

Additional comments

Figure 2 Despite that the approach is interesting, I don’t that the authors employed the results of the figure for anything important. Even less the dendrogram, which is related to culture phase rather than differences. Maybe it can be eliminated.

Figure4B The standard deviations of IN and HA are overlapping?? Differences are significant???. This is an important issue since the authors based part of the discussion on differences in cytotoxic activity, which seems weak in the figure.

Reviewer 2 Report

This group have reported previously the isolation and preliminary characterization of an “isogenic” pair of S.aureus strains isolated at the beginning of mastitis infection and following in-host adaptation as chronic subclinical mastitis developed.  The main genetic difference between the two isolates is the presence of a sigB point mutation in rsbU which affects expression of the stationary phase / stress response sigma factor SigB

The paper catalogues some phenotypic  differences  between the two strains growing under in vivo-like conditions of iron and oxygen limitation. They compared the surface proteome by two methods and few qualitative differences in protein composition were noted. Differences in glycopolymer detected by spectroscopy could not be attributed to differences in capsular polysaccharide or the biofilm associated polysaccharide PNAG so the reason for the difference remains unknown and requires further investigation. The sigB defect is known to result in increased expression of the cytolytic alpha toxin in other strains  and this was confirmed here   in terms of toxin  expression and cytotoxicity towards bovine epithelial cells

This papers demonstrates  the consequences of a sigB mutation on surface polysaccharide and alpha toxin expression in a single bovine adapted strain. It does not show how this difference contributes the chronic mastitis infections especially when alpha toxin is known to be a virulence factor causing tissue destruction in acute mastitis. The paradox between high alpha toxin expression in the host adapted isolated and chronic mastitis needs elaboration.  One hypothesis for prolonged survival in the infected mammary gland is that bacteria can survive intracellularly thus avoiding host defences. High alpha toxin expression is not consistent with maintenance of epithelial cell integrity

Perhaps RPMI does not truly reflect the environment of the infected mammary gland which will be rich in milk proteins and lactose as well as iron restricted and low in oxygen.

The paper reports properties of a single pair of strains isolated from a case of chronic mastitis. Does the sigB mutation occur in many other isolates? How representative is this of adapted strains from other mastitis isolates?

Round 2

Reviewer 1 Report

In this reviewed version, the authors assessed most of my concerns to the better extent possible with the data available.

I still believe that experimental design is extremely weak since they are using a single sample.

In addition, the statement that RPMI is mimicking the mastitis environment of the udder is quite brave from my point of view.

The authors cannot improve the reliability of the data without performing a large quantity of additional experimental procedures, which basically will transform the data into a different thing.

Therefore, I recommend that the authors stated very clearly these facts and the limitations of the study in the manuscript and the title, perhaps stating it in some form to avoid creating expectations that the data does not support.

Reviewer 2 Report

No further comments
